# Quantification of Grapiprant and Its Stability Testing under Changing Environmental Conditions

**DOI:** 10.3390/biomedicines10112821

**Published:** 2022-11-05

**Authors:** Paweł Gumułka, Monika Tarsa, Monika Dąbrowska, Małgorzata Starek

**Affiliations:** 1Department of Inorganic and Analytical Chemistry, Faculty of Pharmacy, Jagiellonian University Medical College, 9 Medyczna St., 30-688 Kraków, Poland; 2Doctorial School of Medical and Health Sciences, Jagiellonian University Medical College, 16 Łazarza St., 31-530 Kraków, Poland; 3Department of Organic Chemistry, Faculty of Pharmacy, Jagiellonian University Medical College, 9 Medyczna St., 30-688 Kraków, Poland

**Keywords:** grapiprant, veterinary drugs, drug quality, osteoarthritis, TLC-densitometry, UPLC-MS/MS, validation of the method

## Abstract

Grapiprant is a new analgesic and anti-inflammatory drug belonging to the piprant class, approved in 2016 by the FDA Veterinary Medicine Center for the treatment of pain and inflammation associated with osteoarthritis in dogs. It acts as a highly selective antagonist of the EP4 receptor, one of the four prostaglandin E2 (PGE2) receptor subtypes. It has been shown to have anti-inflammatory effects in rat models of acute and chronic inflammation and clinical studies in people with osteoarthritis. The current state of knowledge suggests the possibility of using it in oncological therapy. The manuscript presents the development of conditions for the identification and quantitative determination of grapiprant by thin-layer chromatography with densitometric detection. The optimal separation of the substance occurs using silica gel 60F_254_ chromatographic plates and the mobile phase containing ethyl acetate-toluene-butylamine. Validation (according to ICH requirements) showed that the developed method is characterized by straightness of results in a wide concentration range with the limit of detection of 146.65 µg/mL. The %RSD values of the precision and accuracy confirm the sensitivity and reliability of the developed procedure. Next, the method was used for quantification of grapiprant in a pharmaceutical preparation, and for stability studies under various environmental conditions. Additionally, the mass studies were carried out on the stressed samples using the UPLC-MS/MS method. The degradation products were primarily characterized by comparing their mass fragmentation profiles with those of the drug. The results indicated a potential degradation pathway for grapiprant.

## 1. Introduction

Non-steroidal anti-inflammatory drugs (NSAIDs) are considered the most frequently used group of pharmaceuticals, due to their importance in the treatment of many diseases. It is estimated that every day around the world 30–50 million people use a drug from this group [1]. Above all, their anti-inflammatory, analgesic and antipyretic effects are emphasized, however, the incoming reports on the known and new representatives of this group prove a wide range of their therapeutic possibilities. Therefore, although we have several dozen representatives of NSAIDs, new ones are still being looked for, with more and more selective action and minimization of the side effects associated with their use [2]. Although it seems to have been well understood over the years, the mechanism of action of NSAIDs is still being analyzed in detail, and new compounds are being tested in terms of their influence on its various elements.

Prostaglandin E2 (PGE2) is a highly bioactive, endogenous molecule that binds to four G protein-coupled receptors (GPCRs): EP1, EP2, EP3 and EP4. It is produced from arachidonic acid by processes regulated by cyclooxygenases (COX) and prostaglandin synthetases. Due to the fact that it is assessed as the most frequently produced prostanoid in animals, it is of interest to many researchers [3,4]. Under physiological conditions, prostaglandin E2 plays an important role as a regulator of many processes. [5,6]. 

The action of prostaglandin E2 in inflammation deserves special attention. Redness and swelling, resulting from increased blood flow to the tissue caused by vasodilation and increased microvascular permeability, are the effects of PGE2 mediation [4,6,7]. A large number of side effects of classic NSAIDs (e.g., gastrotoxicity) and selective COX-2 inhibitors (e.g., cardiotoxicity) encourage the search for new solutions in the treatment of inflammation [8,9,10]. Therefore, attention has been paid to the possibility of manipulating the processes mediated by PGE2 and exploiting the therapeutic potential of EP receptor modulation. In a mouse model of arthritis, animals lacking the EP4 receptor, compared to those without EP1–3 receptors, showed resistance to symptoms such as swelling or redness [11]. It is worth noting that the amino acid sequence homology of the EP4 receptor between humans and mice is 88% [12]. 

Grapiprant (Table 1) is a new non-inhibitory cyclooxygenase representative of anti-inflammatory drugs [13]. The first information about grapiprant appeared in 2007, when Nakao et al. published work on an extremely strong and selective antagonist of both human and rat EP4 receptors, which are activated by the overarching pro-inflammatory mediator—PGE2 [14]. In March 2016, the drug was approved by the FDA Veterinary Medicine Center for the control of pain and inflammation associated with osteoarthritis in dogs. Currently, it is available in the form of tablets, and the prescribed dose is 2 mg/kg body weight once a day [15,16]. However, it turned out that blocking the EP4 receptor may have benefits in other pathophysiological conditions. Clinical trials are currently underway on the use of grapiprant in humans in the treatment of non-small cell lung cancer and colorectal cancer [17]. Grapiprant (CJ-023423) is a representative of a new class of pharmaceuticals − piprants, defined in 2013 by the World Health Organization. Its mechanism of action points to a new quality in pain management—elblocking a single receptor for the prostanoid responsible for pain and inflammation without interfering with the production and activity of homeostatic prostanoids. 

Previous research on grapiprant indicates its very high potential in the treatment of pain and inflammation. It has been considered safe from a long-term oral administration perspective and has proven to be a good alternative for pain relief in dogs with osteoarthritis (OA) compared to popular NSAIDs [18,19]. The analgesic effect of grapiprant has also been tested in phase II clinical trials in humans with OA. Based on the obtained results, it was found to be effective in relieving symptoms over a treatment period of 4 weeks, administered once or twice daily. The efficacy of grapiprant 100 mg daily is estimated to be equivalent to an oral dose of naproxen 500 mg twice daily [15]. 

Grapiprant, as a relatively new drug, has not yet been compared in sufficient detail with other representatives of NSAIDs. Current research has been limited to its possible use in the treatment of OA in dogs. The Summary of Veterinary Medicinal Product Characteristics does not recommend its use in combination with other anti-inflammatory drugs due to the lack of sufficient research [16]. In 2019, the results of a randomized trial were presented, that included two separate experiments to assess pain control within 24 h following a single oral dose of firocoxib (Previcox^®^) and grapiprant (Galliprant^®^) in an acute canine model of arthritis [20]. In the case of grapiprant, we are dealing with a multidirectional drug with therapeutic application and, importantly, a highly selective mechanism of action, eliminating the classic side effects of NSAIDs, which often lead to discontinuation of therapy. The EP receptor-specific antagonist may have advantages over others, directed against prostaglandin E2. For this reason, intensive research on this compound is extremely important in order to use its therapeutic potential as effectively as possible. 

Current knowledge about the physicochemical properties and analytical methods for the determination of grapiprant is relatively small (www.medchemexpress.com; www.drugbank.ca; accessed on 5 May 2022). Due to the knowledge of PGE2 multidirectional action and the role of the EP4 receptor, the development of its modulators has become the subject of interest in medical chemistry. It can be expected that grapiprant will soon no longer be the only representative of the piprant class. Compounds, that are drug candidates for use in chronic inflammation, solid tumors or migraine have been developed. Ongoing modifications to their structure will lead to an optimal pharmacokinetic profile [21]. The conducted studies have focused on the development of a method for quantification of grapiprant in plasma by high-performance liquid chromatography [22,23,24]. The available studies on the efficacy of grapiprant use indicate no differences in general health, clinical signs or body weight, and show no apparent side effects [25]. This drug may therefore be an alternative to traditional NSAIDs due to its alternative mode of action, but undertaking further research is absolutely justified. 

As mentioned earlier, grapiprant is a new analgesic and anti-inflammatory drug with broad therapeutic potential, requiring further studies. It is important to develop the most optimal analysis conditions for identification and quantitative assay. The purpose of the study was to develop, optimize and validate the conditions for the qualification and quantification analysis of grapiprant in a pharmaceutical preparation using the thin-layer chromatography technique with densitometric detection. Moreover, we decided to analyze the influence of external factors such as pH, temperature and incubation time on the stability of this active substance, taking into account the storage and use conditions, which have a direct impact on the safety of its use. The obtained results allowed us to indicate potential pathways of grapiprant degradation.

## 2. Materials and Methods

### 2.1. Chemicals and Reagents

The used solvents: methanol, n-propanol, isopropanol, glacial acetic acid, ethanol, diethyl ether (POCH, Gliwice, Poland), ethyl acetate, diethylamine, toluene, n-hexane, cyclohexane, chloroform, ammonia 25%, acetone (Chempur, Piekary Śląskie, Poland), butylamine (Sigma-Aldrich, Steinheim, Germany), and acetonitrile, milli q water (Merck, Darmstadt, Germany) were of an analytical grade. pH buffers (2, 7, 8) were purchased from Mettler-Toledo (Greifensee, Switzerland). 

The standard substance grapiprant was purchased from Sigma-Aldrich (SML3183). Pharmaceutical preparation Galliprant (MAH: Elanco GmbH, Cuxhaven, Germany, the manufacturer responsible for batch release: Elanco France S.A.S., Huningue, France), tablets for dogs, was analyzed. Each tablet contains the active substance grapiprant in the amount of 20 mg.

### 2.2. Optimization of Analysis Conditions

Chromatography is an analytical method based on physicochemical phenomena used for the separation, and qualitative and quantitative analysis of mixtures. Among the various chromatographic techniques, TLC is one of the most popular in the pharmaceutical analysis. It is an important method of analysis in many drug trials, especially due to several significant advantages over HPLC or GC, including: simplicity of implementation, high possibilities of components visualization, no requirements for a high purity and concentration of the sample, relatively cheap and easy-to-use equipment, enabling the simultaneous separation and quantification of many samples simultaneously [26,27,28]. Currently, the TLC method is constantly being improved, including by using new adsorbents, building modern equipment and developing new software to optimize the separation.

The first stage of work was to establish and optimize the conditions of the chromatographic analysis by selecting the mobile and stationary phases, which will allow the identification of the test substance, as well as the analysis in the presence of probable degradation products (basic and acidic hydrolysis). Mixtures of different solvents (e.g., toluene, n-propanol, ethyl acetate, cyclohexane, chloroform, methanol, ammonia 25%, butylamine, and glacial acetic acid) in various volume ratios were experimentally tested. For this purpose, prepared solutions (5 μL) were applied successively to chromatographic plates. Then, the plates were developed in chromatographic chambers (Sigma-Aldrich, St. Louis, MO, USA) using selected solvent mixtures, dried at room temperature and analyzed under UV light (254 and 366 nm) to select the developing system optimal for further determinations. Based on the registered densitograms, the appropriate values of the retardation factors R_F_ were established.

The analysis of the obtained results revealed that for several phases, the substance followed the eluent front or remained on the starting line. Finally, it was decided for a study to be conducted using a mixture containing ethyl acetate: toluene: butylamine (2:2:1, *v/v/v*) as eluent. Such conditions are the best way to allow for the analysis of grapiprant in a pharmaceutical preparation and also to enable the visualization of changes taking place in solutions of the active substance under variable environmental conditions.

### 2.3. Chromatographic Analysis

The determination of grapiprant was performed by TLC with densitometric detection. TLC silica gel 60F_254_ plates (1.05554; Merck, Darmstadt, Germany) were used as a stationary phase. On such plates, 5 µL of grapiprant methanolic solutions (0.1%, *w/v*) were applied using an Linomat V sample applicator (CAMAG, Muttenz, Switzerland), with a rate of 300 nL/s. The spots were 5 mm wide with the spacing at 5 mm, and the distance from the side and bottom edges was 10 mm. The mobile phase was a mixture composed of ethyl acetate: toluene: butylamine (2:2:1, *v/v/v*). The chromatographic separation was carried out on a 90 mm path. After development (approx. 30 min), the chromatograms were dried at room temperature, placed in the photo-optical chamber of the densitometer (TLC Scanner 3 with winCATS 4 software version 1.44), and detected at 254 nm.

Qualitative analysis was conducted on the basis of R_F_ values and absorption spectra in the range of 200–400 nm, recorded directly from chromatograms (Figure 1).

For quantitative analysis, the areas of the corresponding peaks recorded on the densitograms were used (Figure 2). 

### 2.4. Validation of the Analytical Method

The next stage of the research was the validation of the developed method. The validation of an analytical method is a process carried out to demonstrate that a procedure is scientifically relevant, credible and reliable, and serves the intended analytical purposes [29]. The examination of certain parameters and the presentation of objective evidence confirms that the requirements for a specific application are met. The purpose of the analytical procedure must be clearly defined as it regulates the validation parameters to be assessed. The main parameters taken into account when validating the chromatographic method are:-accuracy—expresses the relationship between the value considered as standard, and the value resulting from the analysis. According to the ICH guidelines, the accuracy of the method should be assessed on the basis of at least 9 test results for at least 3 different concentration levels, and should be expressed as a percentage of the recovery of a specified amount of standard;-precision—determines the degree of agreement of a series of results of determinations of the same homogeneous sample under certain measurement conditions. The precision of an analytical procedure is usually expressed as the variance, standard deviation or coefficient of variation of a series of measurements at three concentration levels;-repeatability (intra-day)—expresses the precision of determinations carried out under the same conditions (the same analyst, apparatus, reagents), in a short period of time;-intermediate precision (inter-day)—expresses the precision of the determinations made in the laboratory (various analysts, equipment, days, etc.);-specificity—means the ability to determine the analyte in the presence of other components that may be present in the sample, e.g., auxiliary substances, production impurities, degradation products, etc.;-limit of detection (LOD)—the lowest concentration of an analyte in the sample, which can be detected but not necessarily quantified;-limit of determination (LOQ)—the lowest concentration of the analyte in the sample, which can be quantified with appropriate precision and accuracy;-linearity of the analytical method—means the ability (within a specified range) to obtain results directly proportional to the concentration of tested substance. It should be determined by analyzing samples with analyte concentrations covering the declared concentration range of the method and described by a mathematical equation: y = ax + b, where: y—the detector response, x—the concentration of the substance, a—the slope of the calibration curve, b—the intercept. The correlation coefficient r defines the degree of relationship between the variables x and y;-analytical method range—defines the interval between the upper and lower concentrations of the analyte in the sample for which the method is sufficiently precise, accurate and linear.

### 2.5. Determination of the Grapiprant Content in the Pharmaceutical Preparation

The pharmaceutical preparation Galliprant 20 mg (tablets for dogs) was tested, according to the procedure described above. The peak area values obtained on the densitograms were used to determine the active substance content in the preparation. The obtained results along with the statistical evaluation are presented in Table 2.

### 2.6. Stability Study of Grapiprant

The next stage of the research was to analyze the stability of the grapiprant under various pH and temperature conditions, also taking into account the incubation time. For this purpose, 0.2% (*w/v*) solutions of the test substance were prepared with the use of solvents of different pH levels (HCl, NaOH, buffers). Then, the solutions were subjected to various temperatures (room temperature, 70 °C), with sampling for determinations after the incubation time according to the defined test plan. The collected samples were diluted 1:1 (*v/v*) with methanol, and further determinations were carried out according to the procedure described above. The obtained series of chromatograms were subjected to densitometric detection at a wavelength of 254 nm. On the recorded densitograms, additional peaks were present in addition to the main peak from the grapiprant, which probably derived from its degradation product(s). All peaks were well separated and did not interfere with each other, which confirms the possibility of using the developed procedure for grapiprant analysis in the presence of co-present substances.

Regardless of the above procedure, samples of the grapiprant solutions subjected to acidic hydrolysis were analyzed by UPLC-MS/MS, in order to identify the degradation products. They were analyzed using a Waters UPLC-MS/MS system, which consisted of a Waters UPLC (Waters, Milford, MA, USA) coupled to a Waters mass spectrometer (electrospray ionization mode ESI-tandem quadrupole). Chromatographic separation was carried out using the UHPLC Cortecs T3 (2.1 × 100 mm, 1.7 μm) column, maintained at 40 °C. The following gradient of mobile phase was used: 0.1% ammonium formate (FA) in MilliQ water (A)/0.1% FA in acetonitrile (B): 0–10 min/5–50% B, 10–13 min/50–99% B, 13–15 min/99% B, 15–15.01 min/99–5% B, 15.01–20 min/5% B (+5 min 5% B post-time). Chromatograms were recorded using Waters DAD detector, in a 190–500 nm spectrum range. The MS detection settings of the Waters mass spectrometer were as follows: positive ion polarity, capillary voltage 3500 V, nozzle voltage 1000 V, fragmentor voltage 120 V, skimmer 65 V, octopole 750 V, gas temperature 300 °C, gas flow 10 L/min, nebulizer gas pressure 35 pisg, sheath gas temperature 350 °C, sheath gas flow 12 L/min, collision energies—20 and 40 eV. The MS and MS/MS data were obtained in a scan mode ranging from 70 to 1700 *m/z* with a scan rate of 3 spectra/s.

## 3. Results and Discussion

The EP4 receptor, belonging to the GPCRs family, is one of four prostaglandin E2 receptors [3,21]. The EP receptor subtypes: EP1, EP2, EP3 and EP4, although all respond to PGE2, show significant differences in biochemical properties and tissue localization. Studies conducted in mice deficient in each EP receptor subtype determined the direction of PGE2 action, mediated by each of them, and assessed their role in various physiological and pathophysiological responses [4,30,31]. The variety of these processes makes research into EP receptor modulators very intensive. Both agonists and antagonists of prostaglandin E2 receptors have been shown to have wide therapeutic application, and the EP4 receptor is the most promising and versatile in its action. It participates in anti-inflammatory, anticoagulant and vasoprotective reactions, as well as promoting neoplastic or pro-angiogenic processes. The availability of strains of mice with EP4 receptor ablation deepened the understanding of PGE2 as a therapeutically significant mediator and highlighted the validity of efforts to search for selective agonists and antagonists of this receptor as potential drug candidates [12,31]. Numerous publications describing the discoveries of new EP receptor modulators relate to EP4 receptor antagonists. The first one, AH-23848, has been replaced by stronger and more selective ligands [31,32,33,34,35]. Activation of the EP4 receptor can cause a variety of cellular responses such as promoting angiogenesis, proliferation and metastasis, or delaying cancer cells apoptosis. The current state of knowledge suggests a high therapeutic value of selective EP4 receptor antagonists, although further knowledge about them is required [36,37,38,39].

Grapiprant is a highly selective antagonist of the EP4 receptor that mediates nociception induced by prostaglandin E2 [15,40]. The pathological effects of PGE2 are inhibiting by drugs from the group of cyclooxygenase inhibitors. However, their administration causes a reduced production of this prostaglandin, and thus the impairment or even complete elimination of its homeostatic functions. By inhibiting only the EP4 receptor, this disadvantage does not occur [13]. It is effective in treating pain in dogs and may be better tolerated than other NSAIDs. Clinical practice shows that this drug relieves pain (in owner and veterinarian judgment), especially in dogs with osteoarthritis. At the same time, the provided observations confirm its safety, also in older dogs (no differences in general health). Treatments of dogs with OA did not show any toxic effects of grapiprant compared to drugs that inhibit COX, and may therefore offer a more targeted and better tolerated treatment for pain in dogs. Apart from the therapeutic aspect, the quality of the available preparations is a very important factor. The high quality of pharmaceutical preparations and the continuous need for its control are very important for the safety of pharmacotherapy. This study presents a new, validated method of qualitative and quantitative analysis of grapiprant in a pharmaceutical preparation using the TLC technique with densitometric detection. To our knowledge, it also shows, for the first time, a potential degradation pathway for grapiprant. 

During the process of optimizing the assay conditions, a number of parameters were varied, such as the type of stationary phase, the composition of the mobile phase and developing distance. Various ratios of different organic solvents were tested. The variation in the stationary and mobile phases led to significant changes in chromatographic parameters such as peak symmetry and retardation factor. As a result of these experiments, optimal conditions for the analysis of the grapiprant on TLC plates 60F_254_ as the stationary phase and the mixture consisting of ethyl acetate: toluene: butylamine (2:2:1, *v/v/v*) as the mobile phase were created. The obtained chromatograms were subjected to densitometric detection. The selected conditions allowed us to obtain compact spots on the chromatograms and a good peak shape on the densitograms. Additionally, the absorption spectrum of the grapiprant in the wavelength range from 200 to 400 nm was recorded. Based on these observations, a wavelength of 254 nm was selected for drug quantification. The retardation factor (R_F_) determined for the grapiprant in the discussed conditions is 0.35. 

Next, the developed method was validated in order to show its credibility. The linearity, precision, accuracy and robustness of the method were determined. One of the basic parameters to determine is the linearity and range of the method. For this purpose, standard solutions with concentrations from 40 to 2000 μg/mL have been prepared and applied to the plate. After developing, densitometric detection was performed; peak areas corresponding to a specified concentration were registered. The obtained calibration curve confirms a very good fit of the regression line to the actual data. The confidence interval around the regression line includes all points, which confirms the interdependence between the analyzed variables. The equation of the obtained curve is as follows: *p* = 11.91∙c + 2464.43 with the correlation coefficient r = 0.9955. It can be concluded that the model fits very well with the empirical data. The ‘r’ value is very close to unity, it shows the stronger correlation relationship. In addition, other statistical parameters: standard deviation of the slope (S_a_ = 0.46), standard deviation of the intercept (S_b_ = 529.14), standard error of estimate (S_e_ = 841.49) have lower values, so the model fits well. 

Based on the obtained data, the limits of detection (LOD) and quantification (LOQ) were calculated using the formulas: LOD = 3.3∙S_b_/a and LOQ = 10∙S_b_/a, where: a—slope of the calibration curve and S_b_—standard deviation of the intercept. The calculated LOD and LOQ values are 146.65 and 444.39 μg/mL, respectively. These parameters were characterized by relatively low values, which indicates that the developed method is sufficiently sensitive to the analyzed drug. The above values allow us to determine the linearity of the grapiprant quantification method in the range from 444 to 2000 µg/mL with a high correlation coefficient (r = 0.9991; a = 10.76, b = 4107.90, S_a_ = 0.23, S_b_ = 299.62, S_e_ = 276.44), which confirms the high predictive ability of the developed method.

Based on the observed results, the predicted and residual values were analyzed. This enabled the verification of regression assumptions and the detection of outliers. The plot of the residuals model against the independent variable shows that the residuals are irregularly distributed (Figure 3). Thus, it can be assumed that the residuals are random, which means that the assumption of randomness is fulfilled. The designated regression equation and correlation coefficient are as follows: RR (raw residuals) = −0.7∙10^-4^ + 0.13∙10^-6^∙c and r = 0.11∙10^-6^. 

The mean of the residuals in the model is equal to zero, which means that the developed model was not conditioned by single observations, the values of which significantly differ from the predicted ones. Cook’s distance (0.2593), which is a measure of the influence of a given case on the regression equation, was also analyzed. The obtained values shows that none of the considered cases had a significant impact on the loading of the coefficients of the regression equation. Another assumption for the residual values is the lack of autocorrelation of the random component. The autocorrelation was tested using the Durbin–Watson test, where the value of ‘d’ close to 0 indicates the existence of a positive, and close to 4 indicates a negative autocorrelation. A ‘d’ value close to 2 means no autocorrelation. For the considered cases, the test value was d = 0.68, which indicates that the developed regression model has residual values with autocorrelation properties. 

The precision of the proposed method was assessed by analyzing the peak areas obtained for the tested compound on the same day (for intra-day precision) and after a week (for inter-day precision). For this purpose, the prepared 0.01% methanolic grapiprant solution was applied to the plates in the form of bands with a volume of 5 µL each. The plates were analyzed under the conditions described above, and the obtained peak area values were analyzed statistically. In order to determine the intra-day precision, an analogous analysis was performed for freshly prepared solutions, five days apart. The obtained results are presented in the Table 2. The calculated values of the RSD coefficients allow the method to be considered precise.

The accuracy of the method was determined by preparing solutions of the reference substance and the drug, and then applying them to the chromatographic plates in the form of bands: standard substance solution, drug solution and a mixture containing 80 %, 100 % and 120% of the grapiprant substance added to the drug solution, respectively. The obtained peak area values allowed to define the accuracy as a percentage of the recovery of the analyte in the tested sample. The calculated values are collected in Table 2. For the determinations, the percent recovery results were satisfactory, ranging from 98.51 to 101.54% with the %RSD lower than 0.84, which means that the method can be considered accurate. 

The validation report indicates that the developed method fulfills the criteria of an analytical method designated for quantitative control of pharmaceuticals in terms of specificity, linearity, limits of detection and quantification, precision and accuracy. 

Subsequently, the developed method was successfully used to determine the content of grapiprant in a pharmaceutical preparation. The obtained content of the active substance in the tested drug (20.29 mg) shows very good compliance with the amount of 20 mg declared by the manufacturer. This indicates satisfactory accuracy and precision in the analysis of the grapiprant tablet. The obtained results confirm the usefulness of the developed method. 

In the next stage of our work, the stability of the grapiprant under different conditions of pH, temperature and incubation time was investigated. At the stage of optimization of the separation conditions, it was observed that in the acidic environment, as well as under the influence of increased temperature and extended incubation time, grapiprant degrades with the formation of new products (Figure 4 and Figure 5). On the basis of the obtained peak area values at specific measurement time points, it was estimated that the degradation process was much faster at the temperature of 70 °C. Along with the extension of the sample incubation time, a gradual decrease in the peak area of the active substance to 0 was observed, for solutions in 1MHCl, 0.5MHCl, 0.1MHCl and pH 2 buffer. In the case of alkaline or neutral samples, the degradation occurred to a significantly lesser extent. Based on the obtained peak area values, the percentage of the substance was calculated at each measurement time point at room temperature. The results are shown in Figure 6 and summarized in Appendix A. 

Degradation at room temperature was carried out for 63 days (1512 h). The percentage of grapiprant during the measurements was approximate. No markedly accelerated degradation was observed in any of the solutions. All the time, the lowest content of the tested substance was 31.3% in the buffer medium with pH 2, and the highest 54.7% in 0.1MHCl. Comparing the results obtained for room temperature with those obtained at 70 °C, one can notice, first of all, a clear influence of temperature on the degradation process. An analogous analysis was performed for samples incubated at 70 °C. The calculated percentages of substances at each time point are summarized in Appendix A and shown in Figure 7. 

Degradation at 70 °C was carried out for 29 days (696 h). The obtained results indicate a markedly lower stability of the grapiprant in acidic solutions compared to those of basic and neutral character. In the case of using 0.1MHCl as a solvent and a buffer with pH 2, after 16 days (384 h) no test substance was found in the sample. In the case of neutral solutions, the degradation process was similar—the percentages of grapiprant at analogous time points were similar, and after 29 days the content of the tested substance was below 5.5%. For alkaline solutions, this value ranged from about 14% to almost 44% when 0.1MNaOH was used as the solvent. 

Kinetics is an extensive part of the science that studies the movement of bodies. Chemical kinetics is the study of reaction rate, determining the effect of various factors (e.g., pressure and temperature) on the reaction mechanism. By experimentally establishing the relationship between reaction rate and substrate concentrations, a kinetic equation can be empirically determined. The main kinetic parameters determining the course of the reaction are the reaction rate constant k, and half-life t_0.5_ (the time after which the concentration of the substance is reduced by half) [41]. 

The next step of our study was to determine the basic kinetic parameters of the grapiprant degradation reaction. In order to determine the order of the reaction, based on the obtained results of the percentage of substance at specific time points, the dependence curves ln[%]_t_ = −kt + ln[%]_0_ (y = ax + b) were drawn, which indicates that these reactions are in accordance with the 1st order kinetics (Figure 8). Table 3 presents the values of the parameters of the plotted curves and the values of the r coefficient for individual analytes. The values of the reaction rate constants k and the times t_0.5_ and t_0.1_ were also calculated. The obtained results, presented in Table 4, confirm the highest degradation rate in acidic solutions at a temperature of 70 °C. 

The presented studies complement the knowledge of grapiprant with important information, especially from the point of view of the active substance stability. The developed, optimized and validated analytical procedure for the qualitative and quantitative analysis of this substance using TLC meets the needs of searching for analytical methods that are not only highly precise, but also cheap and easy to carry out. It is the first time that TLC in combination with densitometry was reported for the quantification of grapiprant. The developed conditions may therefore turn out to be useful in drug analysis, meeting the requirements for analytical methods used in pharmaceutical analysis. The described method meets the conditions of linearity, precision and accuracy—in accordance with the requirements of ICH, which determines its usefulness and reliability. A high correlation of the results was obtained in a wide range of concentrations. The calculated statistical parameters confirm the high degree of agreement of the results as well as the sensitivity and the accuracy of the method. The degradation process takes place most quickly in an acidic environment (0.1MHCl) at a higher temperature. The analysis of quantitative changes in grapiprant concentration under the influence of various environmental conditions indicates a greater stability of the drug in a slightly alkaline environment and at a lower temperature. The determined kinetic parameters (reaction rate constants k, times t_0.5_ and t_0.1_) of the grapiprant degradation reaction in particular environments confirm the previous observations. 

As mentioned above, degradation of the grapiprant in an acidic environment revealed the presence of several new products. The mass fragmentation pathway of the drug was established using UPLC-MS/MS data acquired in ESI positive ionization mode. The recorded chromatograms, for the standard substance solution and samples after degradation in 0.5MHCl solution, revealed several additional peaks (apart from the main peak from grapiprant; retention time, t_r_ = 8.883 min) with t_r_ = 2.4, 4.36 and 6.06 min (Appendix A). The mass spectra (Appendix A) show that the molecular ion peak was observed at *m/z* 492. Using the elemental composition calculator, the best possible molecular formulas of the fragments were determined. The obtained mass spectra of compounds formed as a result of protonated drug hydrolysis led to the formation of three main product ions: *m/z* 367, *m/z* 323 and *m/z* 295. Their further fragmentation led to the formation of the main ion products at *m/z* 174. Therefore, the probable pathway of grapiprant degradation under the tested conditions may follow the scheme in Figure 9. As mentioned earlier, the recorded densitograms showed the presence of additional peaks next to the grapiprant peak. Comparison of the respective retention coefficients (R_F_ for TLC and t_r_ for UPLC; Figure 4 and Appendix A) suggests that the additional peaks present in the densitogram may come from substance N-{2-[4-(2-ethyl-4,6-dimethyl-1*H*-imidazo [4,5-*c*]pyridin-1-yl)phenyl]ethyl}formamide (R_F_ = 0.72; t_r_ = 6.06 min, *m/z* 323) and from substance 2-[4-(2-ethyl-4,6-dimethyl-1*H*-imidazo[4,5-*c*]pyridin-1-yl)phenyl]ethan-1-amine (R_F_ = 0.86; t_r_ = 2.4 min, *m/z* 295). On the other hand, successive peaks formed during subsequent incubation of grapiprant samples in an acidic environment (shown in Figure 5) indicate further degradation of this substance and/or its degradants.

## 4. Conclusions

Summing up, the presented analytical procedure can be a quick, simple and accurate tool for the determination and stability testing of grapiprant, which may be of importance, inter alia, in the production of a drug, its storage or research on its content in biological material. Moreover, the study of the grapiprant degradation process provided valuable information on the structure of the resulting products and the mechanism of their formation. Due to the fact that the available literature lacks papers on the analytical aspects of grapiprant research, the presented work is a valuable voice in the discussion of news and issues related to veterinary medicine. The possibility of a wide application of the developed procedure allows the dissemination of the acquired knowledge in a wide range of professions, both among scientists and veterinarians. The results can be helpful in developing drug formulations, establishing specifications for potential impurities in the drug substance and medicinal products, and monitoring quality.

## Figures and Tables

**Figure 1 biomedicines-10-02821-f001:**
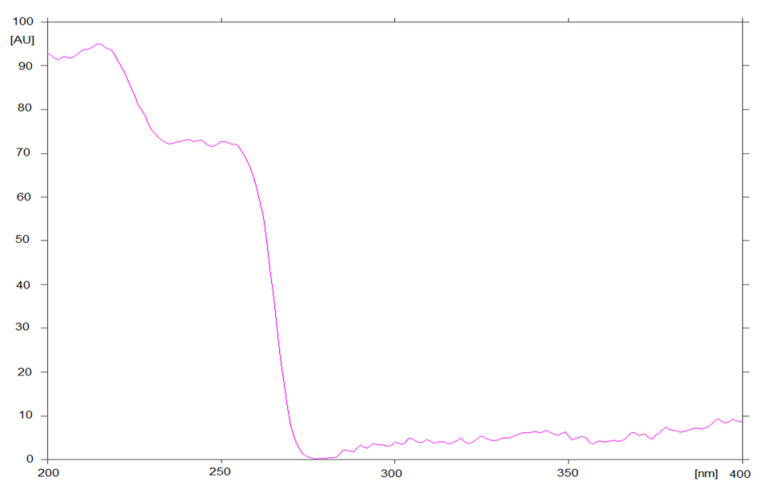
An absorption spectrum of grapiprant, recorded directly from the chromatogram.

**Figure 2 biomedicines-10-02821-f002:**
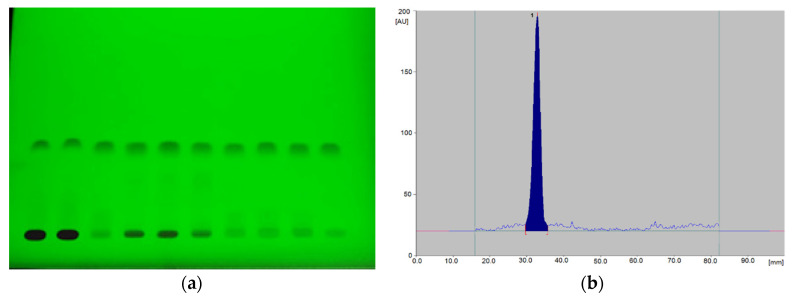
An example of the chromatogram (**a**) and densitogram (**b**), obtained for grapiprant solutions (at 254 nm).

**Figure 3 biomedicines-10-02821-f003:**
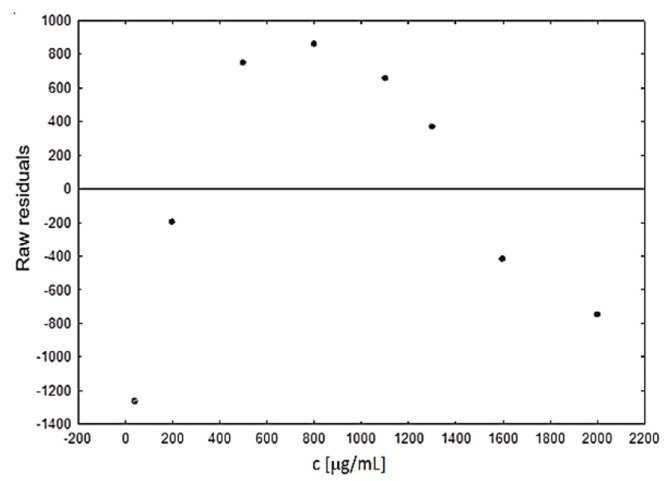
The scatter plot of raw residuals against the predicted values of the independent variable.

**Figure 4 biomedicines-10-02821-f004:**
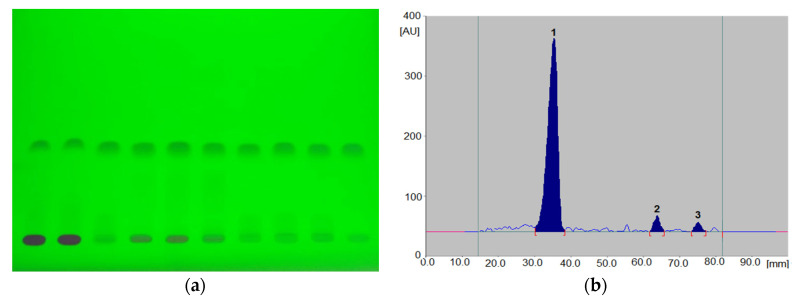
An example of the chromatogram (**a**) and densitogram (**b**; 1-grapiprant, 2,3-degradation products) for grapiprant solutions incubated 24 h at room temperature.

**Figure 5 biomedicines-10-02821-f005:**
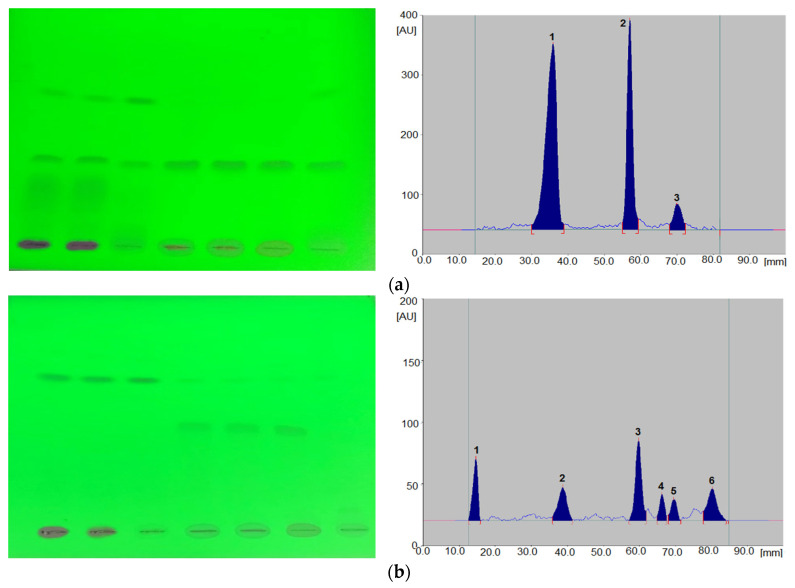
An example of chromatograms and densitograms for solutions incubated: (**a**; 1-grapiprant, 2,3-degradation products) 96 h at 70 °C and (**b**; 2-grapiprant, 1,3,4,5,6-degradation products) 696 h at room temperature.

**Figure 6 biomedicines-10-02821-f006:**
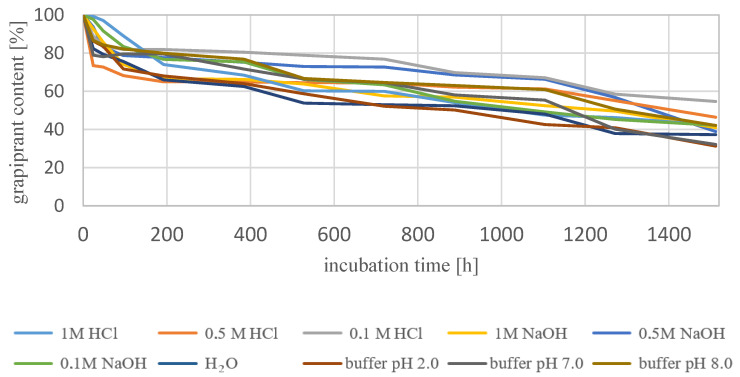
Percentage of grapiprant content during degradation at room temperature as pH function.

**Figure 7 biomedicines-10-02821-f007:**
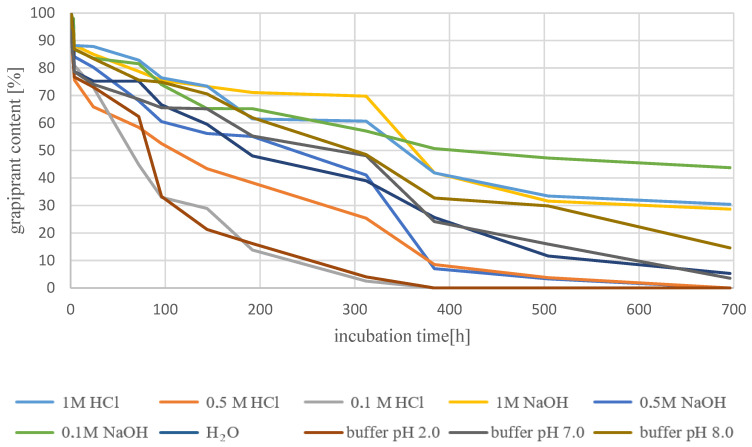
Percentage of grapiprant content during degradation at 70 °C as pH function.

**Figure 8 biomedicines-10-02821-f008:**
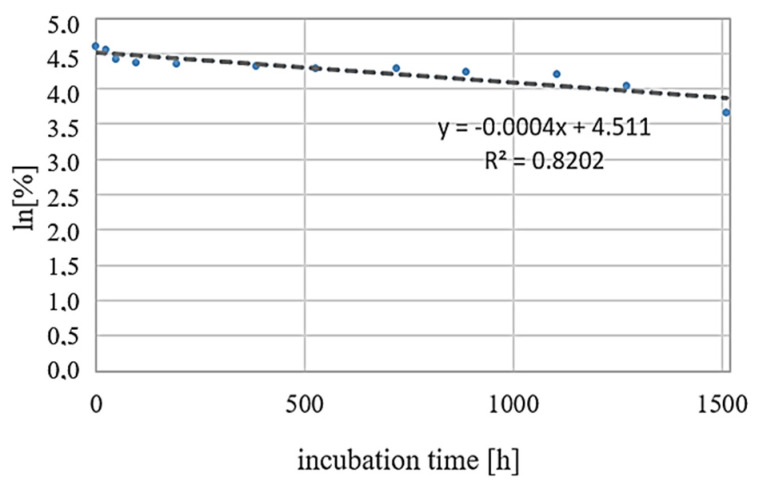
An example of ln[%]_t_=-kt+ln[%]_0_ dependence for grapiprant solution incubated in 1M HCl at room temperature.

**Figure 9 biomedicines-10-02821-f009:**
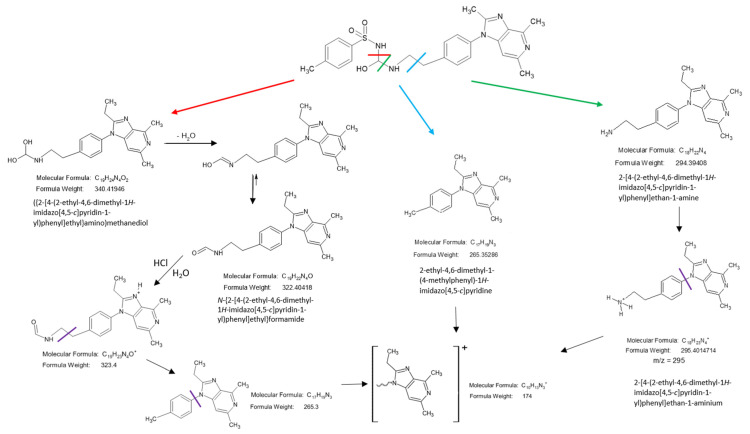
Structures and possible fragmentation pattern of grapiprant.

**Table 1 biomedicines-10-02821-t001:** A structure and general information on grapiprant (methylbenzenesulfonamide).

IUPAC Name	3-[2-(4-{2-ethyl-4,6-dimethyl-1H-imidazo [4,5-c] pyridine-1-yl} phenyl) ethyl]-1-(4-methylbenzenosulfonyl) urea
structure	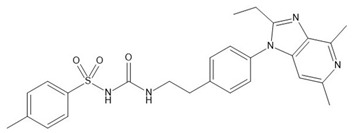
synonyms	CJ-023423; RQ-00000007; AAT-007
molecular formula	C_26_H_29_N_5_O_3_S
molecular weight	491.61 g/mole
CAS number	415903-37-6
Physicochemical properties	
appearance: character, color	white/almost white powder
solubility	DMSO ≥ 50 mg/mL (101.71 mM) methanol—well soluble water—very slightly soluble
melting point	>136 °C (grapiprant hydrochloride)
logP	4.56

**Table 2 biomedicines-10-02821-t002:** Results of method validation and determination of grapiprant in pharmaceutical preparation.

Parameter		Statistical Evaluation
Precision	intra-day	x_m_ = 2306.28 SD = 23.00 %RSD = 1.00
	inter-day	x_m_ = 2606.18 SD = 27.25 %RSD = 1.05
Accuracy	80%	x_m_ = 100.73 SD = 0.85 %RSD = 0.84
	100%	x_m_ = 100.29 SD = 0.78 %RSD = 0.78
	120%	x_m_ = 98.73 SD = 0.79 %RSD = 0.80
Content		x_m_ = 20.29 [mg/tablet] SD = 0.52 %RSD = 2.57

**Table 3 biomedicines-10-02821-t003:** Values of the parameters of the curves ln[%]_t_ = −kt + ln[%]_0_ (acc. y = ax + b) and correlation coefficients (*r*) for solutions incubated at room temperature and 70 °C.

Temperature	Environment	A	b	r
room temp.	1MHCl	−0.0064	4.6950	0.9673
	0.5MHCl	−0.0062	4.5441	0.9866
	0.1MHCl	−0.0080	4.3215	0.9512
	H_2_O	−0.0039	4.5739	0.9871
	0.1MNaOH	−0.0013	4.4978	0.9494
	0.5MNaOH	−0.0018	4.5361	0.9805
	1MNaOH	−0.0018	4.5407	0.9669
	buffer pH 2	−0.0080	4.3558	0.9562
	buffer pH 7	−0.0040	4.5747	0.9616
	buffer pH 8	−0.0025	4.5609	0.9911
70 °C	1MHCl	−0.0004	4.5110	0.9056
	0.5MHCl	−0.0003	4.3410	0.8267
	0.1MHCl	−0.0003	4.5036	0.9529
	H_2_O	−0.0006	4.4021	0.9559
	0.1MNaOH	−0.0006	4.5298	0.9865
	0.5MNaOH	−0.0006	4.5250	0.9699
	1MNaOH	−0.0005	4.4420	0.9445
	buffer pH 2	−0.0006	4.4383	0.9758
	buffer pH 7	−0.0006	4.4849	0.9509
	buffer pH 8	−0.0004	4.4961	0.9689

**Table 4 biomedicines-10-02821-t004:** Values of calculated kinetic parameters of the grapiprant degradation.

Environment	k [1/h]		t_0.5_ [h]		t_0.1_ [h]	
	70 °C	Room Temp.	70 °C	Room Temp.	70 °C	Room Temp.
1MHCl	0.0068	6.24∙10^-4^	101.9	1543.3	15.5	168.8
0.5MHCl	0.0065	5.07∙10^-4^	106.6	1366.9	16.2	207.7
0.1MHCl	0.0118	3.99∙10^-4^	58.7	1736.8	8.9	263.9
H_2_O	0.0042	6.52∙10^-4^	165.0	1062.9	25.1	161.5
0.1MNaOH	0.0012	5.71∙10^-4^	577.5	1213.7	87.8	184.4
0.5MNaOH	0.0018	5.72∙10^-4^	385.0	1211.5	58.5	184.1
1MNaOH	0.0017	5.91∙10^-4^	407.6	1172.6	61.9	178.2
buffer pH 2	0.0103	7.68∙10^-4^	67.3	902.3	10.2	137.1
buffer pH 7	0.0048	7.50∙10^-4^	144.4	924.0	21.9	140.4
buffer pH 8	0.0028	5.72∙10^-4^	247.5	1211.5	37.6	184.1

## Data Availability

Not applicable.

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
