# Peer review of "Quantification of Grapiprant and Its Stability Testing under Changing Environmental Conditions"

_biomedicines, 2022, doi:10.3390/biomedicines10112821_

Round 1

Reviewer 1 Report

In general, figures where TLC chromatograms appear, the X and Y axis labels and numbers should be legible. Besides, it should be appearing full’s peak (without cuts in the Y axis)

Chromatographic peaks should be clearly identified in the figures. According the text, grapiprant peak corresponding to Rf 0.35 but, What products corresponding the showed peaks in figures 4 and 5? Would it be possible to relate each of them with the grapipant's degradation products obtained by MS (figure S9)?

What is the correlation coefficient of the range 440-2000ug/mL? Parameters such as slope and X-axe intercept should be recalculated for this new linear range. Likewise, it should be good to know uncertainty of this both parameters, besides to the sample concentration’ uncertainty

Reviewer 2 Report

The manuscript (biomedicines-1984581) titled "Quantification of grapiprant and its stability testing under changing environmental conditions by TLC-densitometry" is fascinating and within the journal scope, and, in my opinion, suitable for publication. Moreover, it will be interesting for researchers in the same field and the general public. The introduction is too long, and the authors could try to shorten it. The authors should correct a few English issues.
